# Diffusing Wave Microrheology in Polymeric Fluids

**DOI:** 10.3390/polym16101332

**Published:** 2024-05-09

**Authors:** George David Joseph Phillies

**Affiliations:** Department of Physics, Worcester Polytechnic Institute, 100 Institute Road, Worcester, MA 01609, USA; phillies@4liberty.net; Tel.: +1-508-754-1859

**Keywords:** diffusing wave spectroscopy, DWS, microrheology, light scattering, light-scattering spectroscopy

## Abstract

Recently, there has been interest in determining the viscoelastic properties of polymeric liquids and other complex fluids by means of Diffusing Wave Spectroscopy (DWS). In this technique, light-scattering spectroscopy is applied to highly turbid fluids containing optical probe particles. The DWS spectrum is used to infer the time-dependent mean-square displacement and time-dependent diffusion coefficient *D* of the probes. From *D*, values for the storage modulus $G^{\prime }\left(\omega \right)$ and the loss modulus $G^{\prime \prime }\left(\omega \right)$ are obtained. This paper is primarily concerned with the inference of the mean-square displacement from a DWS spectrum. However, in much of the literature, central to the inference that is said to yield *D* is an invocation $g^{\left(1\right)}\left(t\right)=\exp\left(-2q^{2}\bar{X{\left(t\right)}^{2}}\right)$ of the Gaussian Approximation for the field correlation function $g^{\left(1\right)}\left(t\right)$ of the scattered light in terms of the mean-square displacement $\bar{X{\left(t\right)}^{2}}$ of a probe particle during time *t*. Experiment and simulation both show that the Gaussian approximation is invalid for probes in polymeric liquids and other complex fluids. In this paper, we obtain corrections to the Gaussian approximation that will assist in interpreting DWS spectra of probes in polymeric liquids. The corrections reveal that these DWS spectra receive contributions from higher moments $\bar{X{\left(t\right)}^{2n}}$, n>1, of the probe displacement distribution function.

## 1. Introduction

Maret and Wolf [1] and Weitz and Pine [2] proposed a novel path for using light-scattering spectroscopy to study diffusion by mesoscopic probe particles though highly turbid fluids that are in thermal equilibrium. In conventional quasielastic light-scattering spectroscopy (QELSS) studies of probe motion through complex fluids, the probe concentration is sufficiently small that the scattered light is dominated by photons that were scattered once by a probe particle before they emerge from the scattering cell. Diffusing Wave Spectroscopy (DWS) studies complex fluids in which the probe concentration is sufficiently large such that the fluid is opaque so that photons are scattered many times by probe particles before they emerge from the scattering cell.

The individual scattering events experienced by photons in a DWS experiment are no different from the scattering events experienced by photons in a QELSS experiment. In either case, the effect of a single-scattering event on the scattered light is described by the light’s field correlation function $g^{\left(1\right)}\left(t\right)$, whose time dependence is determined by the change in the position $r_{i}\left(t\right)$ of a scattering probe particle *i* during time *t*, namely,
(1)$$g^{\left(1\right)}\left(t\right)=\langle \exp\left(-ıq\cdot \left(r_{i}\left(\tau +t\right)-r_{i}\left(\tau \right)\right)\right)\rangle .$$

Here, q is the scattering vector determined by the laser wavelength and the angle through which the light has been scattered, and the angle brackets 〈⋯〉 denote an average.

In a DWS experiment, each photon experiences many scattering events, not just one. For a single photon in a DWS experiment, scattering events are far enough apart in space that the underlying probe motions are not correlated with each other. The cognomen Diffusing Wave Spectroscopy refers to a particular model for interpreting the field correlation function of the multiple-scattered light. For diffusing probes, standard treatments [1,2] of DWS invoke the Gaussian approximation to assert that $g^{\left(1\right)}\left(t\right)$ determines the mean square of the magnitude of the displacement $X\left(t\right)=r_{i}\left(\tau +t\right)-r_{i}\left(\tau \right))$ of the individual probes, namely,
(2)$$g^{\left(1\right)}\left(t\right)=\exp\left(-q^{2}\bar{X{\left(t\right)}^{2}}/2\right).$$

The term *Gaussian approximation* refers to the mathematical form of the probability distribution P(X,t) that a probe particle will diffuse through a distance *X* during a time *t*. The Gaussian approximation is that P(X,t) has the form of a Gaussian in *X*. The Gaussian approximation arises from Doob’s Theorem [3], which shows that the Gaussian approximation is exact if the displacement X(t) is composed of a large number of small, independent steps, i.e., if X(t) is generated from a Markoff process. However, the assumptions that make the Gaussian approximation correct were shown by Doob to have a second consequence, namely, if X(t) satisfies these assumptions, then it is also mathematically certain that the mean-square value $\bar{X{\left(t\right)}^{2}}$ of X(t) increases linearly in time, namely,
(3)$$\bar{X{\left(t\right)}^{2}}=2Dt,$$
where *D* is a constant that is independent of *t*. In this case, the field correlation function becomes
(4)$$g^{\left(1\right)}\left(t\right)=\exp\left(-q^{2}Dt\right),$$
which is a pure exponential in time.

Many polymer solutions and melts are viscoelastic; probe particles have a memory of past displacements. Correspondingly, in a polymeric fluid, successive small displacements of a probe particle may well be correlated, so probe motion is no longer described by a Markoff process. The requirements for the validity of the Gaussian approximation are no longer obtained. It is appropriate to ask if a concern for the possible validity of the Gaussian approximation is well founded, or whether the concern is purely hypothetical.

First, from Equation (Equation 4), if the Gaussian approximation is correct in the system, then the light-scattering spectrum is mathematically required to be a single exponential. For probe spheres in polymer solutions, non-exponential QELSS spectra are ubiquitous [4,5], contrary to Equation (Equation 4), showing that the Gaussian approximation is typically invalid in these systems.

Second, in some systems, it is possible to apply particle-tracking methods to measure P(X,t) directly. If the Gaussian approximation were valid, P(X,t) would be a Gaussian in *X*. Apgar et al. [6], Tseng and Wirtz [7], Wang et al. [8,9], and Guan et al. [10] all reported that P(X,t) of diffusing probes in their complex fluid systems was clearly not a Gaussian in *X*.

This paper treats the contribution of particle motion and scattering path fluctuations to DWS spectra. It will be shown below that the higher moments $\bar{X{\left(t\right)}^{2n}}$, n>1, of the particle displacement make non-negligible contributions to DWS spectra. These moments are mixed with fluctuations from path to path in the number of scattering events and in the total-square scattering vector. Fluctuations lead to interesting challenges for the interpretation of DWS spectra in terms of particle motions. In the following, Section 2 describes the physical basis for calculating a DWS spectrum. Section 3 calculates the effects of a non-Gaussian P(X,t) on the DWS spectrum; several different variables contribute fluctuation terms. These effects are combined in Section 3.3. Section 4 assembles our results.

## 2. Physical Basis of the Diffusing Wave Spectrum

We first examine general considerations [2,11,12] that permit the calculation of the diffusing wave spectrum. In essence, in a DWS experiment, the light is scattered along many different paths before it reaches the detector. The scattered field incident on the photodetector is the amplitude-weighted coherent sum of the electric fields of the light traveling over all multiply-scattered paths. To describe the light scattered along a single path *P* having *N* scattering events, we take the positions of the *N* scattering particles to be $r_{i}\left(t\right)$ for i∈(1,N). The source is located at $r_{0}$; the detector is located at $r_{N+1}$. Unlike the particle positions, the source and detector positions are independent of time. The light initially has wavevector $k_{0}$; the light scattered from particle *i* to particle i+1 has wavevector $k_{i}\left(t\right)$. The light emerges from the system and proceeds to the detector with wavevector $k_{N}$. The wavevectors $k_{0}$ and $k_{N}$ are the same for every scattering path and at all times. The $k_{i}$ for 1≤i<N are functions of time because they link pairs of particles, and the particles move. Paths differ in the number of scattering particles and the ordered list of particles by which the light is scattered. For the scattering along a specific path *P*, the phase shift is
(5)$$\Delta_{P}\phi \left(t\right)=-k_{0}\cdot r_{0}-\sum_{i=1}^{N}r_{i}\left(t\right)\cdot \left(\;k_{i}\left(t\right)-k_{i-1}\left(t\right)\;\right)+r_{N+1}\cdot k_{N}$$
and the total scattered field is
(6)$$E\left(t\right)=\sum E_{P}\exp\left(i\Delta_{P}\phi \left(t\right)\right).$$

The summation is taken over the list of all allowed paths, a path being an ordered list, of any length greater than zero, of particles in the system, subject to the constraint that no two adjoining elements in a list may refer to the same particle. $E_{P}$ is the scattering amplitude associated with a particular path *P*. The phase shift $\Delta_{P}\phi \left(t\right)$ and scattering amplitude depend on time because the particles move. Because there is translational invariance, the distribution of $\exp(i\Delta_{P}\phi \left(t\right))$ is flat modulo 2π, and the distribution of $E_{P}$ is well behaved. E(t) is therefore the sum of a large number of independent nearly identically distributed random variables, so it is reasonable to infer from the central limit theorem that E(t) has a Gaussian distribution. Because the E(t) have joint Gaussian distributions, the intensity autocorrelation function S(t) is determined by the field correlation function of the multiply scattered light.

This description of multiple scattering reduces to the standard description of single-scattered light if the number of particles along a path is constrained to N=1. By comparison with that description, the $r_{i}\left(t\right)$ are the optical centers of mass of the diffusing particles, the single-particle structure factors arising from internal interference within each particle being contained in $E_{P}$. The standard description of scattering in terms of particle positions continues to be correct if the particles are non-dilute and is the basis for calculating QELSS spectra of concentrated particle suspensions.

The field correlation function depends on the phase shift as
(7)$$g^{\left(1\right)}\left(t\right)\equiv \langle E\left(0\right)E^{*}\left(t\right)\rangle =\langle \sum_{P,P^{\prime }}E_{P}\left(0\right)E_{P^{\prime }}^{*}\left(t\right)\exp\left[i\Delta_{P}\phi \left(t\right)-i\Delta_{P^{\prime }}\phi \left(0\right)\right]\rangle .$$

Because the paths are independent, in the double sum on *P* and $P^{\prime }$, only terms with $P=P^{\prime }$ are, on average, non-vanishing. For those terms, the change between 0 and *t* in the phase is rewritten by applying the following: the definition of Equation (Equation 5) of the phase shift; the requirement that the source and detector locations and the initial and final wavevectors are independent of time; the definition $q_{i}\left(t\right)=k_{i}\left(t\right)-k_{i-1}\left(t\right)$ of the scattering vector; and the addition of 0 in the form $q_{i}\left(0\right)\cdot r_{i}\left(t\right)-q_{i}\left(0\right)\cdot r_{i}\left(t\right)$. Re-ordering the terms gives
(8)$$g^{\left(1\right)}\left(t\right)=\langle \sum_{P}E_{P}\left(0\right)E_{P}^{*}\left(t\right)\exp\left(-i\sum_{i=1}^{N}q_{i}\left(0\right)\cdot \delta r_{i}\left(t\right)-\delta q_{i}\left(t\right)\cdot r_{i}\left(t\right)\right)\rangle ,$$
where $\delta r_{i}\left(t\right)=r_{i}\left(t\right)-r_{i}\left(0\right)$ is the particle displacement, and $\delta q_{i}\left(t\right)=q_{i}\left(t\right)-q_{i}\left(0\right)$ is the change in the scattering vector *i* between times 0 and *t*. The distance between scattering events is generally very large compared to the distances over which particles move before $g^{\left(1\right)}\left(t\right)$ has relaxed to zero, and the $|k_{i}|$ always has the same magnitude, so good approximation $q_{i}$ can only be altered by changing the angle between $k_{i-1}$ and $k_{i}$. In consequence, it is asserted that the $\delta q_{i}\left(t\right)$ term is extremely small relative to the $q_{i}\left(0\right)$ term because $|\delta q_{i}\left(t\right)|/|q_{i}\left(0\right)|$ and $|\delta r_{i}\left(t\right)|/|r_{i}\left(t\right)|$ are of the same order. The scattering amplitudes within the $E_{P}$ only change as the scattering angles in each scattering event change, so, by the same rationale, for each path, $E_{P}\left(t\right)$ is very nearly independent of time over the times of interest. Furthermore, while the total of the scattering vectors along each path must match the difference between the initial and final wavevectors, if the number of scattering events along a path is large, the constraint on the total scattering vector has very little effect on the intermediate scattering vectors.

Under these approximations, the field correlation function reduces to
(9)$$g^{\left(1\right)}\left(t\right)=\langle \sum_{P}|E_{P}\left(0\right){|}^{2}\exp\left(-i\sum_{i=1}^{N}q_{i}\left(0\right)\cdot \delta r_{i}\left(t\right)\right)\rangle ,$$
as shown by Weitz and Pine [2]. The outermost average 〈⋯〉 extends over all particle positions and subsequent displacements. No assumption has thus far been made about interactions between scattering particles.

In order to evaluate this form, Weitz and Pine [2] impose three approximations. Essentially equivalent approximations are imposed by other references [11,12]. Each approximation replaces a fluctuating quantity with an average value. The three approximations are as follows.

Approximation (1): The exponential over the sum of particle displacements can be factorized, namely,
(10)$$\sum_{P}\langle |E_{P}\left(0\right){|}^{2}\exp\left(-i\sum_{i=1}^{N}q_{i}\left(0\right)\cdot \delta r_{i}\left(t\right)\right)\rangle =\sum_{P}|E_{P}\left(0\right){|}^{2}\prod_{i}\langle \exp\left(-iq_{i}\left(0\right)\cdot \delta r_{i}\left(t\right)\right)\rangle ,$$
so that averages over particle displacements can be taken separately over each particle. The factorization is valid if the scattering points *along each path* are dilute so that the particle displacements $\delta r_{1}\left(t\right),\delta r_{2}\left(t\right),\ldots$ are independent because under this condition, the distribution function $P^{\left(N\right)}\left(\delta r_{1}\left(t\right),\delta r_{2}\left(t\right),\ldots \right)$ for the simultaneous displacements of the *N* particles of a path factors into a product of *N* single-particle displacement distributions $P^{\left(1\right)}\left(\delta r_{i}\left(t\right)\right)$, one for each particle. In representative DWS experiments, mean free paths for optical scattering are reported as hundreds of microns [12], while the effective range of the very-long-range interparticle hydrodynamic interactions is some modest multiple of the particle radius or a modest number of microns, so the typical distance between scattering events is indeed far larger than the range over which particles along a given path can influence each other’s motions.

The assertion that the scattering points along each path are dilute does not imply that the scatterers are dilute. If the scattering cross-section of the scatterers is not too large, a given photon will be scattered by a particle, pass through many intervening particles, and finally be scattered by another, distant particle. The scattering particles may themselves be concentrated. The physical requirement of the dilution used in Ref. [2] is that the mean free path between serial scattering events of a given photon is much longer than the range of the interparticle interactions. The motion of each scatterer in a path may very well be influenced by its near neighbors, but those near neighbors are almost never parts of the same path. While two very nearby particles could be involved in one path, this possibility involves a small fraction of the entirety of paths. Indeed, if most paths *P* included pairs of particles that were close enough to each other to interact with each other, the above description of DWS spectra would fail completely: Equation (Equation 9) would not follow from Equation (Equation 8) because the terms of Equation (Equation 8) in $\delta q_{i}\left(t\right)$ would become significant.

Finally, refs. [2,11,12] propose that the average over the individual particle displacements may be approximated as
(11)$$\langle \exp\left(-iq_{i}\left(0\right)\cdot \delta r_{i}\left(t\right)\right)\rangle \approx \exp\left(-q_{i}^{2}\bar{X{\left(t\right)}^{2}}/2\right).$$
where $\bar{X{\left(t\right)}^{2}}=\langle {\left({\hat{q}}_{i}\cdot \delta r_{i}\left(t\right)\right)}^{2}\rangle$. The above is the Gaussian approximation for particle displacements. A variety of rationales for Equation (Equation 11) appear in the literature as discussed below. Pine et al. [12] show how Equation (Equation 11) can be replaced for non-diffusing particles when an alternative form is known a priori.

Approximation (2): Each scattering event has its own scattering angle and corresponding scattering vector $|q_{i}|$, which are approximated via $\langle \exp\left(-q_{i}^{2}\bar{X^{2}}/2\right)\rangle \to \exp(-\bar{q^{2}}\;\bar{X^{2}}/2)$ so that the scattering vector of each scattering event is replaced by a weighted average $\bar{q^{2}}$ of all scattering vectors. Note that each pair $q_{i},q_{i+1}$ of scattering vectors shares a common $k_{i}$ so that $\langle q_{i}\cdot q_{i+1}\rangle \neq 0$. As long as the particle displacements $\delta r_{i}\left(t\right)$ and $\delta r_{i+1}\left(t\right)$ are independent, the cross-correlations in the scattering vectors do not affect the calculation.

Approximation (3): The number *N* of scattering events in a phase factor $\exp\left(-i\sum_{i=1}^{N}q_{i}\left(0\right)\cdot \delta r_{i}\left(t\right)\right)$ is approximated as being entirely determined by the opacity of the medium and the length of each path so that all paths of a given length have exactly the same number of scattering events. Weitz and Pine [2] proposed that light propagation is effectively diffusive; there exists a mean distance $l^{*}$ over which the direction of light propagation decorrelates; the number of scattering events for a path of length *s* is always exactly $N=s/\ell^{*}$; and the distribution of path lengths P(s) between the entrance and exit windows of the scattering cell can be obtained from a diffusive first-crossing problem.

These approximations were [2] combined to predict that the field correlation function for DWS is
(12)$$g_{\mathrm{DWS}}^{\left(1\right)}\left(t\right)\;\propto \int_{0}^{\infty }ds\;P\left(s\right)\exp\left(-k_{0}^{2}\bar{X{\left(t\right)}^{2}}s/\ell^{*}\right),$$
where $k_{0}$ is the wavevector of the original incident light and a mean scattering angle is linked by Ref. [2] to $\ell^{*}$. The model was solved for a suspension of identical particles performing simple Brownian motion, which follow Equation (Equation 3). For a light ray entering a cell at x=0, traveling diffusively through the cell to x=L, and emerging for the first time into the region x≥L, Ref. [2] finds the distribution of path lengths, based on the diffusion equation with appropriate boundary conditions. There are different solutions depending on whether one is uniformly illuminating the laser entrance window, is supplying a point source of light, or is supplying a narrow beam of light that has a Gaussian intensity profile. For example, for the uniform entrance window illumination, Equation (16.39a) of Ref. [2] gives for the DWS spectrum
$$g_{\mathrm{DWS}}^{\left(1\right)}\left(t\right)=\frac{L/\ell^{*}+4/3}{z_{o}/\ell^{*}+2/3}\left\{\sinh\left[\frac{z_{o}}{\ell^{*}}{\left(\frac{6t}{\tau }\right)}^{0.5}\right]+\frac{2}{3}{\left(\frac{6t}{\tau }\right)}^{0.5}\cosh\left[\frac{z_{o}}{\ell^{*}}{\left(\frac{6t}{\tau }\right)}^{0.5}\right]\right\}$$
(13)$${\left\{\left(1+\frac{8t}{3\tau }\right)\sinh\left[\frac{L}{\ell^{*}}{\left(\frac{6t}{\tau }\right)}^{0.5}\right]+\frac{4}{3}{\left(\frac{6t}{\tau }\right)}^{0.5}\cosh\left[\frac{L}{\ell^{*}}{\left(\frac{6t}{\tau }\right)}^{0.5}\right]\right\}}^{-1}$$
where $z_{o}\approx \ell^{*}$ is the distance into the cell at which light motion becomes diffusive, and
(14)$$\tau ={\left(Dk_{o}^{2}\right)}^{-1}$$
is a mean diffusion time.

Even for an underlying simple exponential relaxation, the fluctuation in the total path length from path to path causes the field correlation function for DWS to be quite complicated. If Equation (Equation 11) were correct, then—ignoring the other approximations noted above—the inversion of Equation (Equation 13) at a given *t* could formally obtain from $g_{\mathrm{DWS}}^{\left(1\right)}\left(t\right)$ a value for τ and thence for *D* at that *t*.

## 3. Fluctuation Corrections

Each of the above three approximations replaces the average of a function of a quantity with the function of the average of that quantity. Such replacements neglect the fluctuation in the quantity around its average. However, $g_{\mathrm{DWS}}^{\left(1\right)}\left(t\right)$ is actually a function of the fluctuating quantities. It is non-obvious that fluctuations in its arguments can be neglected. We first consider the separate fluctuations in *X*, $q^{2}$, and *N* and then demonstrate their joint contribution to a DWS spectrum.

### 3.1. Fluctuations in Particle Displacement

To demonstrate the relationship between the single-scattering field correlation function
(15)$$g^{\left(1\right)}\left(q,t\right)=\langle \exp\left(-iq\cdot r_{i}\left(t\right)\right)\rangle$$
and the mean particle displacements $\bar{X{\left(t\right)}^{2n}}$, consider the Taylor series
(16)$$\langle \exp\left(-iq\cdot r_{i}\left(t\right)\right)\rangle =\langle \sum_{n=0}^{\infty }\frac{{\left(-iq\cdot r_{i}\left(t\right)\right)}^{n}}{n!}\rangle .$$

On the right-hand side, the average of the sum is the sum of the averages of the individual terms. By reflection symmetry, averages over odd terms in $r_{i}\left(t\right)$ vanish. In the even terms, components of $r_{i}$ that are orthogonal to q are killed by the scalar product. Without loss of generality, the *x*-coordinate may locally be set so that the surviving component of the displacement lies along the *q* axis, so $q\cdot r_{i}=qx_{i}$, leading to
(17)$$\langle \exp\left(-iq\cdot r_{i}\left(t\right)\right)\rangle =\sum_{m=0}^{\infty }\frac{{\left(-1\right)}^{m}q^{2m}\bar{X{\left(t\right)}^{2m}}}{(2m)!}$$
with $\langle x_{i}^{n}\rangle =\bar{X^{n}}$.

Equation (Equation 17) may be written as a cumulant expansion $\exp(\sum_{n}{\left(-q^{2}\right)}^{n}K_{n}/n!)$, the $K_{n}$ being cumulants [13]. Expansion of the cumulant series as a power series in $q^{2}$ and comparison term by term shows
(18)$$g^{\left(1\right)}\left(q,t\right)=\exp\left[-\left(q^{2}\frac{\bar{X^{2}}}{2}-q^{4}\frac{\left(\bar{X^{4}}-3{\bar{X^{2}}}^{2}\right)}{24}+q^{6}\frac{\left(30{\bar{X^{2}}}^{3}-15\bar{X^{2}}\;\bar{X^{4}}+\bar{X^{6}}\right)}{720}-\ldots \right)\right]$$
with $K_{1}=\bar{X^{2}}/2$, $K_{2}=\left(\bar{X^{4}}-3{\bar{X^{2}}}^{2}\right)/12$, etc., the $K_{n}$ and $\bar{X^{2n}}$ being time-dependent. If the distribution function for X(t) were a Gaussian, then all cumulants above the first would vanish (e.g., $K_{2}=0$), and Equation (Equation 18) would reduce to Equations (Equation 2) and (Equation 11). However, X(t) almost never has a Gaussian distribution in interesting systems.

### 3.2. Fluctuations in the Scattering Vector

For a single-scattering event in which the light is deflected through an angle θ, the magnitude of the scattering vector is
(19)$$q=2k_{0}\sin\left(\theta /2\right),$$
where $k_{0}=2\pi n/\lambda$, with *n* being the index of refraction, and λ the light wavelength in vacuo. Cumulant expansions of spectra depend on powers of $q^{2}$ as
(20)$$q^{2}=2k_{0}^{2}\left(1-\cos\left(\theta \right)\right).$$

For DWS, the $q^{2}$ at the scattering points are independent from each other. All scattering angles are permitted. The average over all scattering angles comes from an intensity-weighted average over all scattering directions. For larger particles, the scattering is weighted by a particle form factor but always encompasses a non-zero range of angles. For small particles, no θ is preferred. However, scattering from small particles is not isotropic because light is a vector field. Scattering from small particles is described by dipole radiation, whose amplitude is proportional to sin(ψ), ψ being the angle between the direction of the scattered light and the direction of the polarization of the incident light.

While it is true that light emerging from a turbid medium is depolarized, turbidity depolarization reflects the presence of many different paths, each of which rotates incident linearly polarized light through a different angle. If linearly polarized light is scattered by a typical small particle or a dielectric sphere (the typical probe) of any size, the scattered light from that one scattering event remains linearly polarized, though with a new polarization vector. In a typical QELSS experiment, the incident light is vertically polarized, the first scattering event ψ is measured from the perpendicular to the scattering plane, and the polarization remains perpendicular to the scattering plane, so sin(ψ)=1. Because in multiple scattering the scattering paths are not confined to a plane perpendicular to the incident polarization axis, in general, sin(ψ)≠1. The factor sin(ψ) arises already in QELSS experiments in which the incident laser polarization lies in the scattering plane. For example, for HH scattering (as opposed to the common VV experiment) from optically isotropic spheres, at $\theta =90^{o}$, the scattering intensity is zero.

For small particles, a simple geometric construction relates θ and ψ. Namely, without loss of generality, we may take the scattering event to be at the origin, the incident light to define the +x-direction with its polarization defining the *z*-axis, and the scattering vector to be in an arbitrary direction not confined to the xy-plane. Defining $\left(\theta_{s},\phi_{s}\right)$ to be the polar angles of the scattering direction relative to $\hat{z}$, the +x-axis lies at $\left(\theta_{x},\phi_{x}\right)=\left(\pi /2,0\right)$ and the angle addition rule gives $\cos\left(\theta \right)=\cos\left(\theta_{s}\right)\cos\left(\theta_{x}\right)-\sin\left(\theta_{s}\right)\sin\left(\theta_{x}\right)\cos\left(\phi_{s}-\phi_{x}\right)$, so one has
(21)$$q^{2}=2k_{0}^{2}\left(1-\sin\left(\theta_{s}\right)\cos\left(\phi_{s}\right)\right).$$

The angle $\theta_{s}$ is not the scattering angle θ of Equation (Equation 20). For the mean-square scattering vector from a single-scattering event in a DWS experiment in which all scattering angles are allowed,
(22)$$\bar{q^{2}}\equiv {\left(4\pi \right)}^{-1}\int d\Omega_{s}\sin\left(\theta_{s}\right)2k_{0}^{2}\left(1-\sin\left(\theta_{s}\right)\cos\left(\phi_{s}\right)\right)=\frac{\pi }{2}k_{0}^{2},$$

The next two moments are $\bar{q^{4}}=\frac{11\pi }{8}k_{0}^{4}$ and $\bar{q^{6}}=\frac{17\pi }{4}k_{0}^{6}$. The corresponding cumulants in a $q^{2}$ expansion are $K_{1}=\pi k_{0}^{2}/2$, $K_{2}≅-3.083k_{0}^{4}$ and $K_{3}≅0.7473k_{0}^{6}$. The second cumulant is not negligible with respect to the first, in the sense that the variance ${\left(|K_{2}/K_{1}^{2}|\right)}^{1/2}$ is ≈1.12. For in-plane scattering in a QELSS experiment, the average of $q^{2}$ over all allowed scattering angles would not be given by Equation (Equation 22). It would instead be proportional to $\int d\theta \sin\left(\theta \right)\sin^{2}\left(\theta /2\right)$. The averages for QELSS and DWS differ because for QELSS, only in-plane scattering arises, while in DWS, out-of-plane scattering events are allowed and important, and because in QELSS with VV scattering, the corresponding polarization weighting factor is unity, while in DWS, the allowed scattering angles are polarization weighted by $\sin(\theta_{s})$. A calculation made in the inadequate scalar-wave approximation would overlook this distinction.

In the standard treatment of photon diffusion in a DWS scattering cell, the path length distribution is computed by envisioning photons as random walkers, and solving the diffusion equation as a first-crossing problem to determine the distribution of path lengths. A path length *s* is approximated as containing precisely $s/\ell^{*}$ steps, the fluctuation in the number of steps arising entirely from differences in the lengths of the various paths. MacKintosh and John [11] presented an extended treatment for the path length distribution, using a diffusion picture and saddle point methods to establish the mean $\bar{N}$ number of scattering events and the mean-square fluctuation in that number as averaged over all path lengths. They treated paths involving few scattering events separately, for which a simple diffusion picture does not accurately yield the distribution of scattering events.

In addition to the fluctuations in *N* arising from fluctuations in *s*, for a path of given *s*, there are also fluctuations in *N* that arise because $\ell^{*}$ is only the average distance between scattering events. While a path of length *s* on average contains $\bar{N(s)}=s/\ell^{*}$ scattering events, for paths of given physical length *s*, there will also be a fluctuation $\langle {\left(\delta N\left(s\right)\right)}^{2}\rangle =\bar{N^{2}\left(s\right)}-{\bar{N(s)}}^{2}$ in the number N(s) of scattering events. Scattering is a rate process linear in path length, so it is governed by Poisson statistics. For paths of a fixed length, the mean-square fluctuation is therefore linear in the number of events.

### 3.3. Joint Fluctuation Effect

For fluctuations within an exponential of a multilinear form, the case here, the effect of the fluctuations is given by the multivariable cumulants. Cumulant expansions are well behaved, and converge under much the same conditions that Taylor series expansions are convergent. A cumulant series is particularly interesting if f(a) is close to exponential in *a* because under that condition, the relaxation is driven by $K_{1}$ and the higher-order $K_{i}$ are often all small. Where do the higher-order cumulants modify the field correlation function for Diffusing Wave Spectroscopy? The variables with interesting fluctuations are the displacement *X*, the mean-square scattering vector $q^{2}$, and the number *N* of scattering events in a scattering path. The quantity being averaged is $\langle \exp(-Nq^{2}\left(\Delta X{\left(t\right)}^{2}\right)\rangle$. Repeated series expansions in *N*, $q^{2}$, and ${\left(\Delta X\right)}^{2}$, through the second cumulant in each variable, using the methods of Ref. [14], lead to
$$g_{\mathrm{DWS}}^{\left(1\right)}\left(t\right)=\langle \exp\left(-\bar{N}\left[\frac{\bar{q^{2}}\;\bar{X{\left(t\right)}^{2}}}{2}+\frac{\bar{q^{4}}\left(\bar{X{\left(t\right)}^{4}}-3{\bar{X{\left(t\right)}^{2}}}^{2}\right)}{24}+\ldots \right]\right.$$
(23)$$\left.+\frac{{\bar{X{\left(t\right)}^{2}}}^{2}}{8}\left[{\bar{N}}^{2}\left(\bar{q^{4}}-{\bar{q^{2}}}^{2}\right)+{\bar{q^{2}}}^{2}\left(\bar{N^{2}}-{\bar{N}}^{2}\right)\right]+\ldots )\right)\rangle .$$

Here, $\bar{N}$ and $\bar{N^{2}}$ are the average and mean-square number of scattering events for all paths. One could also take $\bar{N}$ and $\bar{N^{2}}$ to refer to paths of fixed length *s*, with ${\langle \cdots \rangle }_{s}$ including an average over the path length distribution. On the right-hand side of (Equation 23), the lead term of the exponential is the approximant $\exp(-\langle N\rangle \langle q^{2}\rangle \bar{X{\left(t\right)}^{2}})$ of Maret and Wolf [1]. As with quasielastic light-scattering spectroscopy, higher cumulants of P(X,t) can be significant and contribute to $g_{\mathrm{DWS}}^{\left(1\right)}\left(t\right)$.

Equation (Equation 23) shows only the opening terms of series in the fluctuations in $X^{2}$, $q^{2}$, and *N*. The first line of Equation (Equation 23) reflects particle displacements as captured by an individual single-scattering event and iterated $\bar{N(s)}$ times. The second line reflects the fluctuations from path to path in the total-square scattering vector and the number of scattering events, the fluctuations being $\bar{q^{4}}-{\bar{q^{2}}}^{2}$ and $\bar{N{\left(s\right)}^{2}}-{\bar{N(s)}}^{2}$. The time dependence of $g_{\mathrm{DWS}}^{\left(1\right)}\left(t\right)$ in the above arises from the time dependencies of $\bar{X{\left(t\right)}^{2}}$, ${\bar{X{\left(t\right)}^{2}}}^{2}$, and $\bar{X{\left(t\right)}^{4}}$. The term $\bar{X^{4}}-3{\bar{X^{2}}}^{2}$ (and terms not displayed of the higher order in *X*) reflect the deviation of distribution of particle displacements from a Gaussian. If $q^{2}$ and *N* were non-fluctuating, the second line of Equation (Equation 23) would vanish. Because $q^{2}$ and *N* do fluctuate, $g_{\mathrm{DWS}}^{\left(1\right)}\left(t\right)$ gains additional time-dependent terms, not seen in Equations (Equation 12) and (Equation 18) but appearing as the second line of Equation (Equation 23).

The effect of fluctuations in *N* and $q^{2}$ on DWS spectra, phrased as deviations from Equation (Equation 13), was examined by Durian. Durian [15] reported Monte Carlo simulations for photons making random walks through a scattering slab. He computed the path length, number of scattering events, and sum of the squares of the scattering vectors for each path. These simulations determined fluctuations in the number of scattering events and the total-square scattering vectors, and determined the non-zero effect of these fluctuations on $g_{\mathrm{DWS}}^{\left(1\right)}\left(t\right)$. Durian found that the fluctuation in the total square scattering vector *Y* increased more slowly than linearly with increasing path length *s*. A slower-than-linear increase is expected for a fluctuating quantity and does not imply that the second cumulant of *Y* is negligible for long paths. From Durian’s [15] simulations, an inversion of $g^{\left(1\right)}\left(t\right)$ via Equation (Equation 13) to obtain $\bar{X{\left(t\right)}^{2}}$ has systematic errors because Equation (Equation 13) is inexact. Fluctuations described here and measured by Durian contribute significantly to the field correlation function. Durian demonstrated circumstances under which fluctuations in *N* and $q^{2}$ only have effects of some small size on $g^{\left(1\right)}\left(t\right)$. He [15] explained how Monte Carlo simulations could be used to determine the effect of fluctuations so as to make measurements of particle motion more accurate than those given by Equation (Equation 13).

## 4. Discussion

In this paper, we treat the time dependence of diffusing wave spectra. We demonstrate that the time dependence of DWS spectra arises not only from the mean-square particle displacement $\bar{X{\left(t\right)}^{2}}$ but also from the deviations $\bar{X{\left(t\right)}^{4}}-3{\bar{X{\left(t\right)}^{2}}}^{2}$ from a Gaussian displacement distribution and also from higher powers ${\bar{X{\left(t\right)}^{2}}}^{2}$ of the mean-square displacement. This result does not differ qualitatively from the corresponding result for QELSS spectra, in which the time dependence of $g_{s}^{\left(1\right)}\left(t\right)$ arises not only from $\bar{X{\left(t\right)}^{2}}$ but also from higher powers $\bar{X{\left(t\right)}^{2m}}$ of the mean-square displacement. Just as it is generally wrong to write $\exp(-q^{2}\bar{X{\left(t\right)}^{2}})$ for the general QELSS spectrum, so too it is generally wrong to write $\exp(-\langle N\rangle \langle q^{2}\rangle \bar{X{\left(t\right)}^{2}})$ for the general DWS spectrum. Furthermore, even though fluctuations in $q^{2}$ and *N* depend only slowly on time, the fluctuations couple to the strongly time-dependent $\bar{X{\left(t\right)}^{2n}}$ and thus to the time dependence of the spectrum at short times.

Even in the special case in which the QELSS spectrum is a simple exponential depending only on $\bar{X{\left(t\right)}^{2}}$, the DWS spectrum does not completely simplify. From Equation (Equation 23), even if particle motions are entirely characterized by $\bar{X{\left(t\right)}^{2}}$ so that $\bar{X{\left(t\right)}^{4}}-3{\bar{X{\left(t\right)}^{2}}}^{2}$ and similar higher-order terms all vanish, the fluctuations in *N* and $q^{2}$ cause the DWS spectrum to depend on ${\bar{X{\left(t\right)}^{2}}}^{2}$ and higher-order terms. However, in this case, the additional terms found here would lead to rapidly decaying decays in the DWS spectrum; these might be difficult to resolve experimentally. Durian [15] discussed using computer simulations to incorporate these issues and interpret DWS spectra.

Comparison of DWS and macroscopic rheological determinations of the storage and loss modulus has, in some cases, found reasonable agreement [16], suggesting that non-Gaussian diffusion issues are not very important. However, under modern conditions, it is possible to perform orthodox rheological measurements over microscopic rather than macroscopic distance scales. In these true microrheological measurements, a known force is applied to nanometer- or micron-size objects, the resulting motions being used to infer rheological properties. Note, for example, studies of magnetically driven beads [17], probes in an ultracentrifuge [18,19], and polystyrene spheres undergoing capillary electrophoresis through a neutral polymer support medium [20]. These studies found that the storage and loss moduli revealed by microscopic probes in complex fluids are not equal to the storage and loss moduli measured with classical macroscopic rheological instruments. An agreement between DWS and macroscopic rheological measurements therefore does not demonstrate that DWS works because it might be the case that the nominal moduli inferred from DWS measurements do not agree with the moduli measured obtained from orthodox microscopic instruments.

In defense of the original paper of Maret and Wolf [1], the diffusing particles being studied were indeed identical Brownian particles in a simple Newtonian liquid, which is the case for which the Gaussian approximation is correct. Furthermore, Maret and Wolf explicitly identified the Gaussian form of Equation (Equation 2) as a first cumulant approximation, not an exact result.

Finally, one might ask if representative studies of probe motion using DWS give indications that the non-Gaussian issues raised here may be significant. Fortunately, Doob’s Theorem gives a test for the validity of the Gaussian approximation, namely, if the approximation is valid, the probe mean-square displacement increases linearly with time, and the corresponding inferred diffusion coefficient is a time-independent constant. Huh and Furst [21] reported on polystyrene sphere probes in solutions of filimentous actin. In some systems, the mean-square displacement was found to increase linearly in time, so the issues noted here substantially did not arise. In other systems, they reported that $\langle |X\left(t\right){|}^{2}\rangle$ increased as $t^{\alpha }$ for α≈3/4, in agreement with direct measurements of $\langle |X\left(t\right){|}^{2}\rangle$ with particle tracking. In these other systems, the issues raised here are potentially significant. Sarmiento-Gomez et al. [22] used DWS to study probe motion in slightly-connected polymer networks, inferring that $\langle |X\left(t\right){|}^{2}\rangle$ in their systems does not increase linearly with *t*, so the issues raised here may be significant. Dennis et al. [23] used DWS to infer the short-time diffusivity of charged silica particles in non-dilute solutions. At short times, the DWS spectrum corresponds to the initial relaxation of $\langle |X\left(t\right){|}^{2}\rangle$, corresponding to the first cumulant of a QELSS spectrum, avoiding the issues revealed here. Dennis et al. found that the concentration dependence of their *D* agreed well with a theoretical model.

## Data Availability

Data are contained within the article.

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
