# Peer review of "Diffusing Wave Microrheology in Polymeric Fluids"

_polymers, 2024, doi:10.3390/polym16101332_

Round 1
Reviewer 1 Report
Comments and Suggestions for Authors
Para 69, correct the word “typicallyinvalid”
Para 78-83. The sentence “In some though----studies need further consideration” is too long. Which need to simply. Also, there some grammatical mistakes in the manuscript, which needs to be addressed, like the phrases ‘studies needs’ in the above sentence should be corrected as ‘studies need’
In the abstract it is mentioned that from D, values for the storage modulus G′(ω) and the loss modulus G′′(ω) are obtained”. However, I couldn’t find any such discussion regarding the storage modulus and the loss modulus in the results& discussion part of the manuscript.
What are the steady-state conditions applied by the authors while applying the Gaussian Approximation in the study
Overall the manuscript needs more clarity in the Results and Discussions part.
Author Response
Referee 1 Response:
I thank the referee for catching a typographic error, and did a further re-read looking for more of these.
Para 78: As the referee requested, I rewrote the paragraph, splitting it into more sentences, and moved the paragraph almost to the end of the paper.
With respect to "studies needs" the relevant clause is "suggesting that the significance of agreement between DWS and macroscopic rheometric studies needs further consideration." In this clause, the verb is "needs", which is singular not plural. However, "studies" is not the subject of the sentence. The subject of the sentence is "significance", which is also singular, so the verb and subject do agree.
With respect to obtaining G'(w) and G"(w), the actual process is outside the topic of the paper. I mention the moduli in the abstract, because the mention explains why the rest of the abstract is of interest. I added to the abstract a sentence emphasizing the paper's actual topic, namely "This paper is primarily concerned with the inference of the mean-square displacement from a DWS spectrum."
The referee asks "what are the steady-state conditions..." Apologies, but I am not sure what the referee has in mind here, though his question sounds important. I added to the first sentence of the paper the clause "that are in thermal equilibrium" which may answer his question. If not, a longer referee explanation would be appreciated.
I added some additional paragraphs in the Discussion which may improve the paper.
Reviewer 2 Report
Comments and Suggestions for Authors
The work by Prof. Phillies revisits the determination of physical properties of polymeric liquids from the diffusing wave spectroscopy (DWS). In particular, most attention is paid for the determination of the diffusion coefficient D. The author convincingly explains the reasons why the D cannot be obtained from the mean-square displacement of a probe particle. Instead, the higher moments of the displacement must be taken into account which is shown by the corrections of the Gaussian approximation interpetating DWS spectra.
I believe that this work should be accepted for publication as it will surve a nice tutorial for future researches in the field. Before the final acceptanle, I only ask to consider a couple of comments:
- Could the author provide an example of some most recent works (for instance, that have been published in the last 5 years) where the neglecting of the higher forms gave, to his opinion, some questionalbe or not so reliable results?
-Line 211. I think it would be better simply to refer to eq.3 instead of writing the eq. 13 since the latter is identical to a former one.
Author Response
Referee 2 response
I thank the referee for his kind words.
With respect to adding works, I have added three references, showing a wide spread of year of publication, discussing how this paper impinges on their findings. In one paper, my issues sometimes might matter and sometimes might not. In a second, my issues consistently might matter. In the third paper, Dennis, et al., found a clever way to avoid the issues that I raise.
Line 211: Done.